# RNR-R2 Upregulation by a Short Non-Coding Viral Transcript

**DOI:** 10.3390/biom11121822

**Published:** 2021-12-03

**Authors:** Karin Broennimann, Inna Ricardo-Lax, Julia Adler, Eleftherios Michailidis, Ype P. de Jong, Nina Reuven, Yosef Shaul

**Affiliations:** 1Department of Molecular Genetics, Weizmann Institute of Science, Rehovot 76100, Israel; karin.broennimann@weizmann.ac.il (K.B.); iricardola@mail.rockefeller.edu (I.R.-L.); julia.adler@weizmann.ac.il (J.A.); nina.reuven@weizmann.ac.il (N.R.); 2Laboratory of Virology and Infectious Disease, Rockefeller University, New York, NY 10065, USA; Eleftherios.Michailidis@rockefeller.edu (E.M.); ydj2001@med.cornell.edu (Y.P.d.J.); 3Division of Gastroenterology and Hepatology, Weill Cornell Medicine, New York, NY 10065, USA

**Keywords:** deoxynucleotides and DNA viruses, RNR-R2 regulation, non-coding RNA, Hepatitis B virus

## Abstract

DNA viruses require dNTPs for replication and have developed different strategies to increase intracellular dNTP pools. Hepatitis B virus (HBV) infects non-dividing cells in which dNTPs are scarce and the question is how viral replication takes place. Previously we reported that the virus induces the DNA damage response (DDR) pathway culminating in RNR-R2 expression and the generation of an active RNR holoenzyme, the key regulator of dNTP levels, leading to an increase in dNTPs. How the virus induces DDR and RNR-R2 upregulation is not completely known. The viral HBx open reading frame (ORF) was believed to trigger this pathway. Unexpectedly, however, we report here that the production of HBx protein is dispensable. We found that a small conserved region of 125 bases within the HBx ORF is sufficient to upregulate RNR-R2 expression in growth-arrested HepG2 cells and primary human hepatocytes. The observed HBV mRNA embedded regulatory element is named ERE. ERE in isolation is sufficient to activate the ATR-Chk1-E2F1-RNR-R2 DDR pathway. These findings demonstrate a non-coding function of HBV transcripts to support its propagation in non-cycling cells.

## 1. Introduction

Replication of DNA viruses requires deoxyribonucleotide triphosphates (dNTPs), the levels of which are tightly regulated in cells. Certain DNA viruses induce cell proliferation upon infection, a process that leads to Ribonucleotide reductase (RNR) upregulation and dNTP synthesis. RNR is a heterodimeric tetramer of R1 and R2 subunits. R2 is encoded by the *RRM2* gene, which is exclusively expressed at the entry to S phase of the cell cycle. Other DNA viruses induce RNR via activation of the cellular DNA-damage response (DDR) pathway. The human papillomavirus (HPV) upregulates RNR-R2 expression by activating the ATR-Chk1-E2F1 pathway [1,2]. Epstein–Barr virus (EBV) and Kaposi’s sarcoma-associated herpesvirus (KSHV) also activate DDR [3,4]. In all these cases, a viral protein was identified to be responsible for DDR activation.

RNR-R2 activity and dNTP synthesis is critical for HBV replication. We have previously shown that inhibition of RNR by HU blocks HBV progeny formation [5,6,7,8,9]. Depletion of intracellular dNTP pools by SAMHD1 is used as a restriction mechanism of viral DNA synthesis, including restriction of HBV progeny formation [10,11,12], demonstrating the importance of RNR upregulation and dNTP levels for viruses.

We have previously reported that HBV upregulates RNR-R2 expression in non-cycling cells by inducing DDR and identified essential steps involved in the process [5,6]. In cells infected with HBV, the Chk1 S/T kinase is activated and in turn activates the E2F1 transcription factor, a key regulator of RNR-R2 transcription [13]. We identified the HBV HBx open reading frame (ORF) to be essential in this process.

Here we report that the HBx protein is dispensable, but rather a small region of the HBx RNA, that we termed embedded regulatory element (ERE), is sufficient to induce the ATR-Chk1-E2F1-RNR-R2 DDR pathway. These findings demonstrate the multitasking role of a viral transcript, functioning as both a coding and non-coding genetic element in reprogramming host gene expression.

## 2. Materials and Methods

### 2.1. Tissue Culture, Treatments and Reagents

HepG2, HEK293, and HEK293T (ATCC) cells were cultured in Dulbecco’s modified Eagle’s medium (Gibco, Thermo Fisher, Waltham, MA, USA) supplemented with 8% fetal bovine serum (Gibco) and 100 U/mL of penicillin and 100 µg/mL of streptomycin (Biological Industries, Sartorius, Göttingen, Germany). To obtain quiescent HepG2, the medium was supplemented with 2% dimethyl sulfoxide (DMSO) for at least six days. The reagents used were UCN-01 (7-hydroxystaurosporine), Caffeine and Doxycycline (Sigma-Aldrich, Merck, Burlington, MA, USA), KU55933 and AZD6738 (ApexBio, Houston, TX, USA). For cell irradiation a XRAD 320 by Precision X-Ray was used. Percent of live cells was calculated by dye exclusion test. Briefly, cell suspension was mixed 1:1 with 0.4 Trypan blue dye (Thermo-Fisher, Waltham, MA, USA), live (unstained) and dead (stained) cells were counted using Countess automated cell counter (Thermo Fisher).

### 2.2. Preparation of HBV

HBV was produced from HepDE19 cells (generously provided by Dr. Haitao Guo, Indiana University) as previously described [14,15,16]. 

### 2.3. Preparation of Lentiviral Transducing Particles and Transduction

A list of the plasmids used is included in the Appendix A, including plasmids described previously [17,18]. Lentiviruses were produced as described [5] using the calcium phosphate method to transfect HEK293T. Lentivirion containing medium was filtered through 0.45 µM cellulose acetate filter, and supplemented with 8 µg/mL polybrene. Virion-containing medium was used to transduce the HepG2 cells. Twelve to twenty-four hours after infection the cells were washed five times in phosphate buffered saline (PBS) and fresh medium was added to the cells. Transduced cells were harvested after 72 h.

### 2.4. Primary Human Hepatocytes

For lentivirus production, 293T-LentiX cells were plated on poly-L-lysine coated plates. The next day, cells were co-transfected with specific pLenti4 plasmids, together with VSV-G and Gag-Pol expressing packaging plasmids, using PEI transfection reagent. Medium was collected after 48 h, and cleared by filtering through a 0.45 µM low protein-binding filter and centrifugation at 4000 rpm for 10 min. Virus was concentrated by ultracentrifugation using a SW41 Ti rotor (Beckman Coulter, Brea, CA, USA) at 20,000 rpm for 1.5 h at 4 °C, and resuspended in hepatocyte defined medium (HDM; Corning, Corning, New York, USA).

Primary human hepatocytes (Lonza, Basel, Switzerland) were transplanted into FNRG mice to create human liver chimeric mice and livers from highly humanized mice were harvested as described [19] and seeded on collagen-coated plates in HDM. Lentivirus transduction was performed overnight in the presence of 8µg/mL polybrene. Cells were harvested after 72 h and RNA was purified using RNeasy kit (Qiagen, Venlo, The Netherlands), including an on-column DNAse I treatment (Qiagen). cDNA was generated as described below.

### 2.5. RNA and Protein Extraction and Analysis

RNA was extracted using TRI Reagent (MRC, Cincinnati, OH, USA), PerfectPure RNA Purification kit (5 PRIME, Qiagen) or RNeasy (Qiagen) for PHH. First-strand synthesis was performed using qScript cDNA synthesis kit (Quanta, Beverly, MA, USA). qRT-PCR was performed using the LightCycler480 (Roche, Basel, Switzerland), with PerfeCta^®^ SYBR Green FastMix mix (Quanta). All qPCRs were normalized to 18S mRNA levels or RPS11 (PHH). The primer sequences are in the Appendix A.

Lysates were prepared from cells using RIPA buffer [20] supplemented with Dithiothreitol (DTT) and protease and phosphatase inhibitors (Sigma-Aldrich), and subjected to SDS-PAGE. Antibodies are listed in the supplements. We used enhanced chemiluminescence (ECL) detection using EZ-ECL (Biological Industries). For quantification of Western blot bands we used ImageJ (version 1.51k) software. (http://imagej.nih.gov/ij, accessed on 12 November 2018). Phos-Tag™-AAL-107 was purchased from Wako Pure Chemical Industries, Ltd. (Osaka, Japan), and was added at 40 µM concentration to 6% SDS-PAGE gel.

### 2.6. Extracellular and Intracellular DNA

Intracellular DNA was extracted as described [16]. Extracellular viral DNA was extracted using Qiaquick minelute virus kit (Qiagen, Düsseldorf, Germany). Quantification of viral DNA was done by qPCR as previously described [16]. Briefly, the qPCR was performed using a TaqMan Universal PCR Master Mix (Applied Biosystems, Foster City, CA, USA). PCR was done using a Roche LightCycler 480 PCR machine and the HBV copies/sample were calculated based on a standard curve composed from 2xHBV plasmid in a concentration range of 109–101 copies.

### 2.7. HBsAg Chemiluminescence Immunoassay

For quantitative analysis of HBsAg in the medium of infected cells, 50 μL of supernatants were loaded into 96-well plates of a chemiluminescence immunoassay (CLIA) kit according to the manufacturer’s instructions (Autobio Diagnostics Co., Zhengzhou, China). Plates were read using a FLUOstar Omega luminometer. Concentrations are expressed in national clinical units per milliliter (NCU/mL).

### 2.8. Liquid Chromatography/Mass Spectrometry

Quiescent HepG2 cells transduced with ERE or mock were harvested by trypsinization and washed in PBS. Solid phase extraction was performed as described [21]. Samples were extracted in 10 mM ammonium acetate and 5 mM ammonium bicarbonate, pH 7.7 and methanol (50:50) buffer with 5 uL of 1 mM C13ATP added for LC-MSMS analysis (Ibid). The MRM parameters for dNTPs were 468 > 112.2 and 468 > 192.1, collision energy CE = 12 for dCTP; 483 > 81, collision energy CE = 27 for TTP; 492.1 > 81 and 492.1 > 136.3, collision energy CE = 24 and 16, respectively, for dATP; 507.9 > 141.1, collision energy CE = 35 for dGTP; and 519.1 > 141.1, collision energy CE = 35 for 13C10-ATP. The residual pellet containing precipitated proteins was used to normalize samples.

### 2.9. RNA-Seq Analysis

The data was downloaded from GSE93153 [21]. Sequencing reads were aligned as previously described [22]. The reads were aligned to the human (hg19) genome and to the HBV genome (GenBank U95551.1). Alignment was performed using Bowtie v.1.1.2 [23] with a maximum of two mismatches per read.

### 2.10. Statistical Analysis

Error bars refer to standard error of the mean (SEM). A two-sided Student’s *t*-test was performed to assess significance.

Data analysis was performed with a web-tool for plotting box plots http://shiny.chemgrid.org/boxplotr/ (accessed on 28 November 2021) and Microsoft Excel.

## 3. Results

### 3.1. RNR-R2 Induction by Non-Coding HBx ORF Mutants

Clustered HBV RNA fragments are detected at the C-terminus of the HBx ORF four hours post-infection of PHH [21]. Already at that time, RNR-R2 is upregulated (Figure 1A). Since it is unlikely that during this short period HBx is synthesized, we examined the possibility that the RNA fragments are active in RNR-R2 upregulation. In agreement with our previous reports [5,6], HBx ORF is as effective as the 1.3× HBV genome in upregulating RNR-R2 expression (Appendix A). To investigate the importance of the HBx synthesis in this process, we inserted several stop codons along the HBx ORF (Figure 1B). Interestingly, we found that all the HBx nonsense mutants were active in upregulating RNR-R2 expression (Figure 1C). Although all the mutants were well expressed at the RNA level (Figure 1D), none produced HBx protein to a detectable level (Figure 1E). Further analysis revealed that neither the deletion of the 3′UTR region, nor the 5′ sequences, including the HA-tag, are involved in RNR-R2 activation (Appendix A). To uncouple the HBx ORF and the RNA sequence, we constructed a synthetic HBx ORF with saturated silent mutations that was inactive in upregulating RNR-R2 (Appendix A). These data suggest that RNR-R2 is upregulated in the absence of HBx protein but depends on the integrity of the HBx RNA sequence.

### 3.2. A Short RNA Region of the HBx ORF Activates RNR-R2

We constructed and analyzed a series of HBx ORF sub-fragments. Remarkably, an internal 125 base-pairs long fragment corresponding to nucleotides 1572-1697 (taking the unique HBV EcoRI site as 1) was sufficient to upregulate RNR-R2 mRNA to the level obtained by the full-length HBx ORF (Figure 2A–C). The short RNA expressed from this fragment was named embedded regulatory element (ERE). The immediate ERE upstream sequence (nucleotides number 1463-1571) was marginally active and used as a negative control RNA in the subsequent experiments. HBx and ERE but not the control RNA increased R2 protein levels (Figure 2D). ERE is functional when expressed from the positive strand under a polymerase II promoter (Appendix A). On average, the ERE fragment leads to about fivefold higher RNR-R2 expression than the negative control RNA (Figure 2E).

We have previously reported that HBV induces RNR-R2 over twofold in infected PHH [6]. To show whether ERE is active under physiological conditions, we transduced PHH and revealed that expression of WT HBx or ERE were sufficient to increase the level of RNR-R2 expression (Figure 2F,G).

Next, we asked whether RNR-R2 upregulation gives rise to higher dNTPs levels. We quantified the dNTP levels in quiescent cells expressing ERE and found that all the four dNTPs were increased in the ERE expressing cells compared to the control (Figure 2H). Overall, these data suggest that ERE upregulates RNR-R2 and increases the cellular dNTP pools.

### 3.3. ERE Induced the ATR-Chk1-E2F1-RNR-R2 Axis

Previously, we reported that HBx induces RNR-R2 in quiescent cells by activating the Chk1-E2F1-RNR-R2 pathway [6]. To demonstrate whether ERE functions by activating this pathway, we used Chk1 inhibitor UCN-01 (Figure 3A) and Chk1 knockdown (Figure 3B and Appendix A) and revealed that both markedly abolished ERE-mediated RNR-R2 upregulation. Under these conditions ERE expression was not compromised (Appendix A). Further, Chk1 was highly phosphorylated at the S345 position in the presence of ERE (Figure 3C). These results suggest that ERE induces Chk1 phosphorylation. In addition, the E2F1 level, the transcription activator of the *RRM2* gene, was increased and phosphorylated under this condition (Figure 3D and Appendix A). The importance of the activation of this axis in HBV infection of PHH was tested by inhibition of RNR-R2 with HU and inhibition of Chk1 with UCN-01. Both HU and UCN-01 treatment led to a significant reduction in HBV RNA, extracellular and intracellular DNA and HBsAg (Appendix A). This was not due to toxicity, as cell viability was not decreased with HU or UCN-01 treatment (Appendix A).

The most upstream kinases transducing DDR signaling are ATM and ATR [25]. The main kinase responsible for Chk1 S345 phosphorylation and activation is ATR [26,27,28]. We tested the possibility of ATR involvement in ERE-induced RNR-R2 upregulation. Caffeine, an inhibitor of both ATM and ATR [29], significantly inhibited RNR-R2 induction, while ERE was well expressed (Figure 3E and Appendix A). However, a specific ATM inhibitor, KU55933 [29], was not effective (Appendix A). Inhibition of induction of p21, a known target of ATM [29], was used as a positive control for KU55933 activity (Appendix A). In contrast, when using a specific ATR inhibitor, AZD6738 [29], RNR-R2 induction by ERE was significantly reduced (Figure 3E), while ERE was well expressed (Appendix A). Similarly, silencing of ATR led to a significant reduction of ERE-mediated RNR-R2 induction (Figure 3F and Appendix A). In addition, ATR phosphorylation at the Thr1989 residue, a marker for ATR activation [30,31], was evident in the presence of ERE (Figure 3G,H). These data support a model that the non-coding ERE RNA fragment activates RNR-R2 gene expression via the ATR-Chk1-E2F1 DDR axis.

## 4. Discussion

We addressed the question of how HBV ensures optimal dNTP levels for its DNA synthesis in non-cycling cells. Initially, we observed that the HBV-positive HepG2.2.15 cells produce large amounts of thymidine [5], indicating a role of HBV in the biosynthesis of dNTPs. Follow-up studies revealed that HBV induces RNR-R2 expression in non-cycling cells via the activation of the Chk1-E2F1-RNR-R2 DDR circuit [6]. We report here that a non-coding 125 bases RNA fragment (ERE) embedded inside the HBx ORF induces the expression of RNR-R2, a cell cycle gene, to increase dNTP levels. We show here that ERE specifically activates the DDR cascade by ATR, but not by ATM, the other main kinase in DDR, hinting at a very specifically acting mechanism.

The ERE sequence is highly conserved and shared by all the viral transcripts. ERE is localized in the HBx ORF, overlapping with the essential DR2 element for viral replication [32,33,34,35]. These overlapping activities provide a double lock mechanism in sequence conservation. Given the strategic position of ERE in the viral genome, we cannot specifically eliminate ERE function by mutagenesis to evaluate its role in HBV life cycle.

Our analysis of the RNA-seq data by Niu et al. [21] of HBV infected PHH revealed that R2 is upregulated early on in infection. Data from the same study detected ERE RNA in early stages of HBV infection (Appendix A). The origin of these RNAs remains to be investigated. Interestingly, an internal HBx transcription start site (TSS) has been reported (around nucleotide 1500) that could potentially generate ERE positive transcripts [36,37]. Whether this TSS is designed to generate an ERE specific transcript is an open question.

The 3.2kbp genome of HBV is very small with a very limited number of ORFs. The question is how such a limited number of genes is sufficient to exploit the cellular pathways and recruit essential cellular machinery to ensure productive infection. Intriguingly, cryptic elements inside the HBV transcripts fulfill some of the tasks. These include the pgRNA packaging signal [38,39], the PRE regulating the nuclear export of the HBV transcripts [40,41,42], the RNA region involved in destabilizing HBV RNA in response to cytokine treatments [43,44] and the ERE described here. Along with the minimization of the HBV genome and the number of ORFs, non-coding RNAs became an effective alternative. The unique feature of HBV genome organization is that the non-coding RNA function is embedded within the mRNA species. Whether a similar mode of combined coding-non-coding RNA is prevalent in other organisms is an interesting and open question to be investigated. 

## Figures and Tables

**Figure 1 biomolecules-11-01822-f001:**
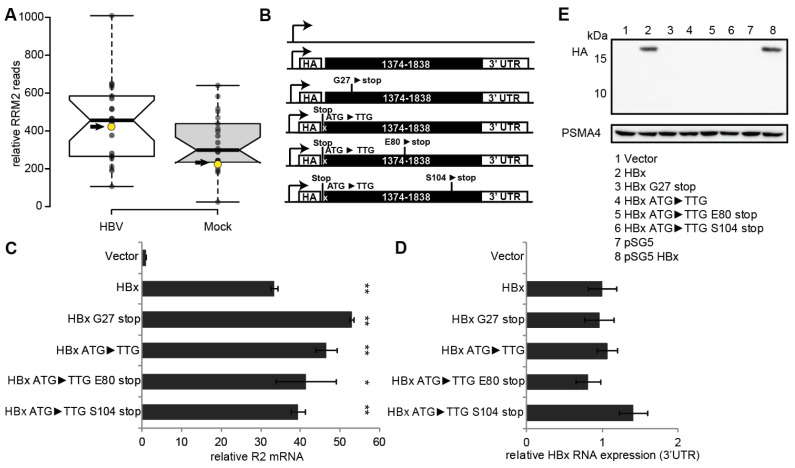
Non-coding HBx ORF transcripts elicit RNR-R2 upregulation in non-cycling cells. (**A**) Boxplot analysis of RNA-Seq data from GSE93153 uploaded by Niu et al. [21] with help of the BoxPlotR web-tool (http://shiny.chemgrid.org/boxplotr/ accessed on 29 November 2021). R2 (*RRM2*) GSEM reads/total GSEM reads *10^7^ of each time point measured were plotted for HBV infected or mock infected PHH. The arrow marks the 4h post infection time point. Center lines show the medians; box limits indicate the 25th and 75th percentiles as determined by R software; whiskers extend 1.5 times the interquartile range from the 25th and 75th percentiles, outliers are represented by dots. *n* = 22 sample points. We performed Student’s *t*-test on the both normalized sample sets and the *p*-value is 0.05689. (**B**) Schematic depiction of the HA-tag-HBx ORF (black regions) with the inserted mutations, all cloned into a Lenti-vector (LV). The HBV 3′UTR was added as a common RNA element for normalization. An empty lenti-vector was used as negative control. (**C**,**D**) Non-cycling HepG2 cells were transduced with the HBx gene mutants of panel B and relative level of the expressed R2 RNA (**C**) and lentiviral construct expression (3′UTR) (**D**) were quantified by qRT-PCR. (**E**) The constructs in (**B**) were transfected into HEK293 cells. HA-HBx protein levels were measured by Western blot with anti-HA antibody. PSMA4 was used as a loading control. The pSG5-empty (Stratagene) and pSG5-HA-HBx constructs [24] were used as negative and positive controls respectively. *p*-value * < 0.05 and ** < 0.01 was calculated for each sample compared to the vector control using Student’s *t*-test, non-labeled samples are non-significant.

**Figure 2 biomolecules-11-01822-f002:**
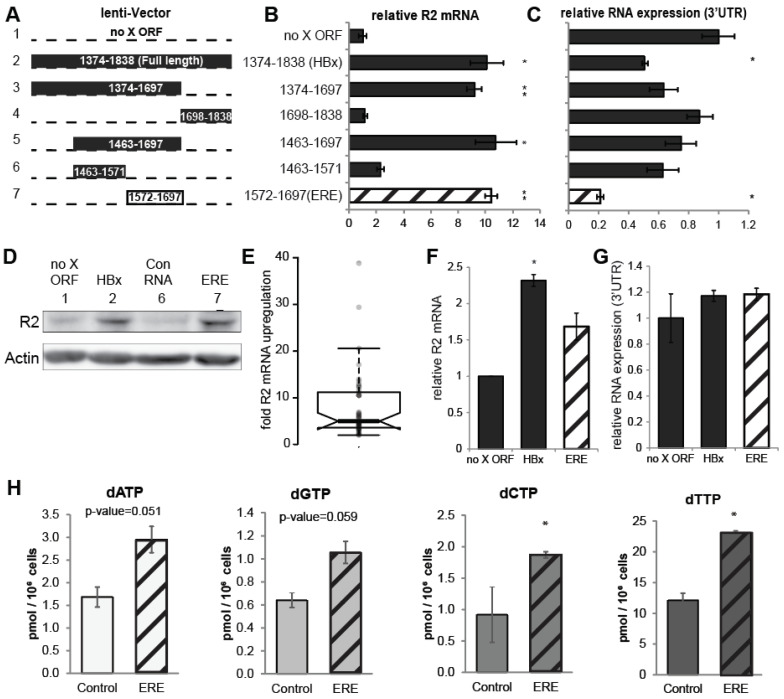
Delineation of the HBx RNA minimal sequence required for RNR-R2 upregulation. (**A**) Schematic representation of the truncated HBx gene in LV constructs. The black regions were cloned for expression. The standard numbering from the EcoRI site in the HBV genome is indicated. As in Figure 1B, the constructs are framed by an HA-tag 5′ and the HBx endogenous 3′UTR. (**B**,**C**) The constructs in panel A were transduced into non-cycling HepG2 cells. Relative RNR-R2 mRNA (**B**) and 3′UTR (**C**) levels were measured by qRT-PCR from three biological replicates. * *p*-value < 0.05, ** *p*-value < 0.01 were calculated for each sample compared to vector control using Student’s *t*-test. (**D**) As in (**B**), Vector, 1.3x HBV or constructs 1, 2, 6, and 7 depicted in A were transduced into quiescent HepG2 cells and Western blot analysis with antibody for R2 was performed. Actin was used as loading control. (**E**) Boxplot of the fold changes of RNR-R2 expression between ERE and the control RNA of 45 biological replicates is shown. Student’s *t*-test was performed, and the results are highly significant (*p* < 0.0001). The boxplot was created with help of the BoxPlotR web-tool (http://shiny.chemgrid.org/boxplotr/ accessed on 28 November 2021). (**F**) Primary human hepatocytes (PHH) were transduced with LV HBx or ERE or no HBx ORF. Relative RNR-R2 mRNA levels of biological duplicates were measured by qRT-PCR. Student’s *t*-test was performed showing a *p*-value of <0.05 for the HBx sample. ERE was non-significant due to the low number of samples. (**G**) As in F, LV RNA levels (3′UTR) were measured by qRT-PCR to assure similar expression of the constructs. The LV expression levels did not change significantly. (**H**) dNTP levels in quiescent HepG2 cells transduced with ERE or control from three biological replicates were measured by LC/MS. The levels for each dNTP are shown and Student’s *t*-test was performed. * *p*-value < 0.05, non-labeled samples are non-significant.

**Figure 3 biomolecules-11-01822-f003:**
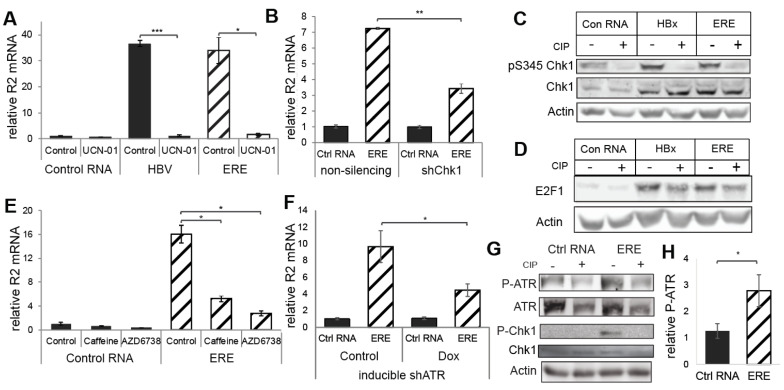
ERE activates the ATR-Chk1-E2F1 DNA damage response axis. (**A**) Non-cycling HepG2 cells were transduced with the indicated constructs and treated with 1 µM UCN-01, an inhibitor of Chk1 kinase activity, for 24 h. Relative RNR-R2 mRNA levels were measured by qRT-PCR from three biological replicates. (**B**) Non-cycling HepG2 cells, expressing shChk1 or non-silencing shRNA were transduced with control RNA or ERE. Relative RNR-R2 mRNA levels were measured by qRT-PCR from three biological replicates. Student’s *t*-test was performed, * *p*-value < 0.05, ** *p*-value < 0.01, and *** *p*-value < 0.001. The *p*-value for the non-silencing is 0.00001, the *p*-value for shChk1 is 0.02090 and the *p*-value between ERE non-silencing and shChk1 is 0.008217. (**C**,**D**) Quiescent HepG2 cells were transduced with LV containing the indicated constructs, control RNA, full length HBV, and ERE. Phospho-S345 Chk1 and Chk1 (**C**) and E2F1 (**D**) levels were measured by Western blot, using a specific antibody, and the phosphorylated nature of the bands was validated by Calf Intestinal Phosphatase (CIP) treatment. Actin protein levels were measured as control. (**E**) Non-cycling HepG2 cells were transduced with LV-constructs containing ERE or control RNA. After 48 h, cells were treated with 2 µM Caffeine or 10 µM AZD6738, a specific ATR inhibitor, or left untreated. RNR-R2 mRNA levels were measured 24h after treatment with qRT-PCR from three biological replicates. Student’s *t*-test was performed comparing the control of each sample to the treatments. * *p*-value < 0.05 (**F**) A HepG2-based cell line that expresses ATR shRNA under a Doxycycline (Dox)-inducible promoter was created. These cells were DMSO treated to be quiescent, transduced with ERE or control RNA and treated with 1ug/mL Dox for 3 days. R2 mRNA levels from three biological replicates were measured by qRT-PCR. * *p*-value < 0.05 (**G**) PhosphoT1989 ATR, PhosphoS345 Chk1, Chk1, and total ATR levels were examined in the presence and absence of ERE by Western blot. CIP was used to evaluate the specificity of the phospho-antibodies. Actin was used as loading control. **H**) Quantification of P-ATR levels of five biological replicates of experiment described in G. * *p*-value < 0.05.

## Data Availability

The RNA-seq data were downloaded from Gene Expression Omnibus accession number GSE93153.

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
