# Peer review of "RNR-R2 Upregulation by a Short Non-Coding Viral Transcript"

_biomolecules, 2021, doi:10.3390/biom11121822_

Round 1
Reviewer 1 Report
The authors found HBx sequence (ERE motif) alone contributed to the HBV-induced upregulation of R2 expression, which depended on ATR-Chk1-E2F1 DDR pathway.
The study was interesting, but the major conclusion that the HBx ERE RNA sequence is responsible for R2 induction is not consolidated enough, the role of ERE DNA sequence in the R2 induction cannot be excluded. Although lentiviruses deliver RNA genome to recipient cells, cDNA will also be generated, and it is still possible that DNA sequence contributes to the R2 induction, even though there was already some clue in Suppl Fig.4C,D that DNA sequence may not be a major driver. However, the authors still need to provide direct evidence that it’s RNA but not DNA sequence that contributes to R2 induction. For instance, full-length HBx or ERE RNA can be in vitro transcribed and directly transfected to HepG2 or PHHs, the induction of R2 is then measured.
Materials and Methods part is not well organized. Please provide the info about the lentiviral vector, cloning and mutagenesis details. The lentivirus preparation should be summarized in a separate section. LC-MS section should contain more details on data requisition, selection, and analaysis.
Please include the statistical analysis result for Fig.1A if possible.
Details on each construct in Fig.1B should be introduced more clearly. For instance, what’s the meaning of “stop” in the 3rd, 4th, and 5th constructs?
In Fig.1E, some names of the constructs were not consistent with those in Fig.1C. For example, the 4th one is HBx ATGATG>TTG E80 Stop, and it was HBx ATG>TTG E80 stop in Fig.1C. Same issue for the 5th one. Please explain what the 6th, 7th, and 8th ones were. No information about pSG5 was seen in Materials except here.
Please provide the Western blot result in Suppl Fig.3 to show the HBx protein was comparably expressed and detected after synonymous mutation.
Please explain why the 3UTR sequence can still be detected in suppl Fig.4E, given that only reversed 3UTR sequence was transcribed.
It’s hard to believe there is no statistical significance between ERE and control RNA in shChk1 cells in Fig.3B, as there was a 3-fold increase, and no big scale bar was there. Please indicate which group was used to normalize the data in Fig.3, and double-check the statistical analysis of Fig.3B.
Panels A, B, and C in suppl Fig.7 were not in the right order based on the legend description.
Reviewer 2 Report
In this study, Broennimann et al. identify a RNA element (ERE) within the HBx open reading-frame of Hepatitis B virus (HBV) that is sufficient to upregulate RNR-R2 expression, a key enzyme for regulating the dNTP pool (by converting NTPs to dNTPs). In previous work, they had identified the upregulation of R2 by the HBx ORF (refs 5,6). In this work, they make the unexpected discovery that this is due to RNA—the ERE element—rather than the HBx protein. This is a new and exciting discovery, which should be of broad interest not only in virology but also in the field of non-coding RNA research.
The data convincingly shows that ERE leads to upregulation of RNR-R2, leading to increased dNTP levels in quiescent HepG2 cells, and that this effect is mediated specifically by ATR-Chk1-E2F1, using specific chemical inhibitors. Overall, the paper is well-written, and the results are clear and supported by the evidence presented. The supplementary data strongly supports these conclusions and this exciting result.
Minor suggestions to further improve the paper would be:
(1) One of the initial observations was clustered HBV RNA fragments at the C-terminus of the HBx ORF (line 138 and ref 19). It would be very helpful for the readers to have a small figure (probably supplemental) displaying this, for example as a single genome track of RNA-seq coverage over this part of the HBV genome (akin to a simplified version of Fig 5 in ref 19). This will also allow to compare the location of this region to the location of ERE; as it seems that the 1572-1697 fragment indeed roughly corresponds to the left side of the RNA-seq peak (on Fig 5 of ref 19).
(2) The size of the functional element within ERE is unknown; the authors test the 1572-1697 fragment (125nt) but it is possible that the actual functional RNA element within this region is smaller. Have the authors tested any further smaller construction? While this is not necessary for this paper, it would be at least important to mention this possibility, especially if the authors intend to further define the “minimal ERE”.
(3) Functional RNA elements typically display secondary structure. It would be easy and informative if the authors add a supplementary figure testing the secondary structure of ERE. Furthermore, if that structure is conserved across related viruses, this would also constitute a very strong evidence for functionality, and may define a new viral non-coding RNA class.
(A quick check using RNAfold indeed seems to indicate a very strong 28nt hairpin roughly in the middle of the 125nt element, potentially folded within a larger, less well-defined hairpin).
(4) In Fig 2E, it would be informative to have not only the boxplot plotted, but also the individual datapoints.
(5) In Fig 3B, the comparison between Ctrl RNA and ERE in shChk1 condition is labeled as “n.s.”, but still appears quite clearly (most likely because there is residual Chk1 activity in the shChk1 cells). What is the exact p-value? Arguably, the comparison between ERE non-silencing and ERE shChk1 could be also added (similarly to Fig 3F).
Minor typos:
page 1 line 40: “is as a restriction mechanism” – the authors probably meant either “is used as a restriction mechanism” or “is a restriction mechanism”.
Reviewer 3 Report
HBV infects human hepatocytes and causes severe diseases. Previously, the authors found that HBV infection induces the DNA damage response pathway and RNR-R2 expression. In the work, the authors further investigated how HBV increases RNR-R2 expression and identified an HBV mRNA embedded regulatory element (ERE) as a non-coding viral transcript that upregulates RNR-R2 expression. The ERE element likely activates the ATR-chk1-E2F1-RNR-R2 DDR pathway. The experiments are well designed and that data are convincing.
I only have a few comments.
- Statistical analysis is missing in some figures, e.g., Figures 1D, S1C, 2C, 2F, 2G and etc.. In Figure 2H, statistical analysis should be performed and labeled using same method.
- In Figure 1E, sample 6 should be labeled as “HBx ATG â–² TTG ”.
- Figures 1A and 2E should show the data points of control samples, other than showing fold change numbers.
- 2F & 2G are not conclusive without proper statistical analysis.
- In Figures S2B, S2C, and S5B, empty vector controls are required.
